# FEDGSE:GRADIENT-BASED SUB-MODEL EXTRACTION FOR RESOURCE-CONSTRAINED FEDERATED LEARNING

## ABSTRACT

Federated Learning with Model Heterogeneity has emerged as an important domain, especially with the increasing number of devices that possess diverse resources. However, many clients with valuable data are unable to contribute to training the global model due to the limitations of their resource-constrained devices. One method to overcome this challenge is to extract sub-models from the global model specifically for these resource-limited clients. Unfortunately, existing methods for sub-model extraction rely on predetermined rules, which fail to consider the relationship between the update gradients of the global and client models. In this paper, we propose a novel method called `FedGSE`, which selects neurons within each layer that exhibit large gradients generated by training the global model on public dataset on the server side, and the selected neurons are used to form sub-models for training on the client side using local dataset. This ensure the gradient updates produced by the sub-model closely resemble the gradient updates that would be produced when training the client data on the global model. As a result, the performance of the sub-model becomes more aligned with that of the global model. Experimental results demonstrate that our method achieves state-of-the-art performance on multiple datasets when compared to other baseline methods.

## 1 INTRODUCTION

Huge data are generated by edge devices such as IoT and sensors(Lim et al., 2020), which can be utilized for training machine learning models. some of this data contains private information and users do not hope to send them to central cloud. To protect data privacy, this data need to be trained locally. Federated Learning (FL)(McMahan et al., 2017; Karimireddy et al., 2020) is a distributed machine learning paradigm, which enables clients collaboratively train global model locally without leaking data information. In recent years, the emergence of Larger Models(Chang et al., 2023) has demonstrated their powerful utility. Machine learning models tend to become larger. As machine learning models become increasingly larger, many edge devices are unable to handle the resource requirements, such as memory and computing capacity, needed to train the entire model(Cai et al., 2020; Lee et al., 2019; Gobieski et al., 2019).

To address these challenges, the concept of extracting a sub-model from the entire model is proposed to address aforementioned issues. Neuron pruning is an efficient method to extract sub-model, which selects neurons layer by layer from entire model base on the edge device capacity. Various methods have been proposed to design strategies for selecting neurons.

The neuron pruning can be divided static distribution and dynamic distribution. The static utilize predetermined rules such as HeteroFL(Diao et al., 2020), which allocate fixed pruning neurons to edge device according to their capacity. However, this method cannot be applied when the size of the global model exceeds the capacity of any individual client's model. The dynamic method addresses this shortcoming by selecting neurons in a continuously changing manner. One approach is Federated Dropout(Caldas et al., 2018), which randomly drops out neurons. But, this method leads to uneven training of neurons. Then FedRolex(Alam et al., 2022) were proposed to overcome this problem, which selects neurons in a rolling way for each client, ensuring every neurons are trained

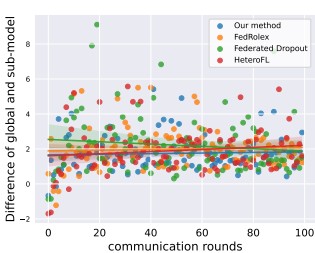
(a) Difference scattered distribution.

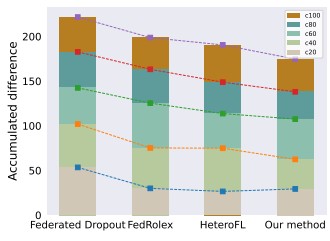
(b) Accumulated difference.

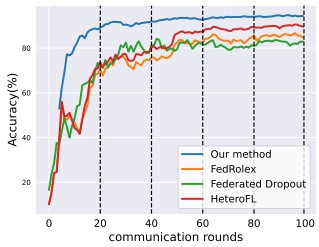
(c) The accuracy of four methods.

Figure 1: (a) The scatter plot and regression curve of parameter difference. (b) The cumulative errors of parameter difference from different communication rounds using different methods. (c) The accuracy of different methods. Data generated under EMNIST.

evenly. These methods primarily focus on designing allocation strategies to ensure training work. However, no one has designed a strategy that starts from the optimization direction of approaching the effectiveness of the global model when extracting sub-model. For this purpose, we conduct the following experiments. In Figure1 we can observe that the parameters of difference between the global model(Simulating training local datasets of clients using the global model from the current communication round to obtain parameters) and sub-model has a significant impact on the model accuracy (as show in Figure1(c)). The smaller the difference, the better the sub-model approximates the effectiveness of the global model. Specifically, in Figure1(b) we can get the **A**ccumulated **D**ifferences (Ad) $\{Ad_{Federated\ Dropout} > Ad_{FedRolex} > Ad_{HeteroFL} > Ad_{\texttt{FedGSE}}\}$ in communication round $\{20, 40, 60, 80, 100\}$. And in Figure1(c), we can obtain the corresponding accuracy rates that are generally $\{Acc_{Federated\ Dropout} < Acc_{FedRolex} < Acc_{HeteroFL} < Acc_{\texttt{FedGSE}}\}$ every above communication round.

Based on this finding, our paper propose gradient-based strategy to select the neurons. As show in Figure2, the first step is to obtain a sub-public dataset that is similar to the client, then global model computes the gradients of every layer's neurons using sub-public dataset. Next, our method selects some neurons within each layer that have relatively large gradient values to form a sub-model, which is then given to client for training. This ensures the gradient updates of this sub-model are closest to the gradient updates of the global model. The core idea of `FedGSE` is extract neurons from the entire model based on their gradients, where neurons with relatively large values are selected layer by layer. Figure1(b) demonstrates that this method effectively reduces the parameters gap between the global and sub-model, resulting in the highest accuracy, as shown in Figure1(c). Our contributions can be summarized as:

- Our method is the first to propose improving Federated Learning training by approximating the update gradients between the global model and sub-models.

- We have mathematically proven that the algorithm we designed in this way can make the update gradients between the global model and sub-models closest.

- To validate the efficiency of the proposed method, we compare state-of-the-art methods. Evaluation results show that `FedGSE` outperforms other comparison methods.

## 2   RELATED WORKS

Many pruning strategies have been proposed to optimize sub-mode extraction progression. These methods train the global model by updating the weights of the extracted sub-model base the predetermined rules. The predetermined rule method can be categorized into two main types in accordance with whether the pruning is dynamic. We named static methods and dynamic methods. In principle, the latter resolved the shortcomings of the former.

**Static Allocation Strategy.** This allocation strategy is based on predetermined rules. The clients can get different fixed pruning neurons sub-model from entire model according to the capacity of clients. They have developed different templates for each resource-constrained client, which are

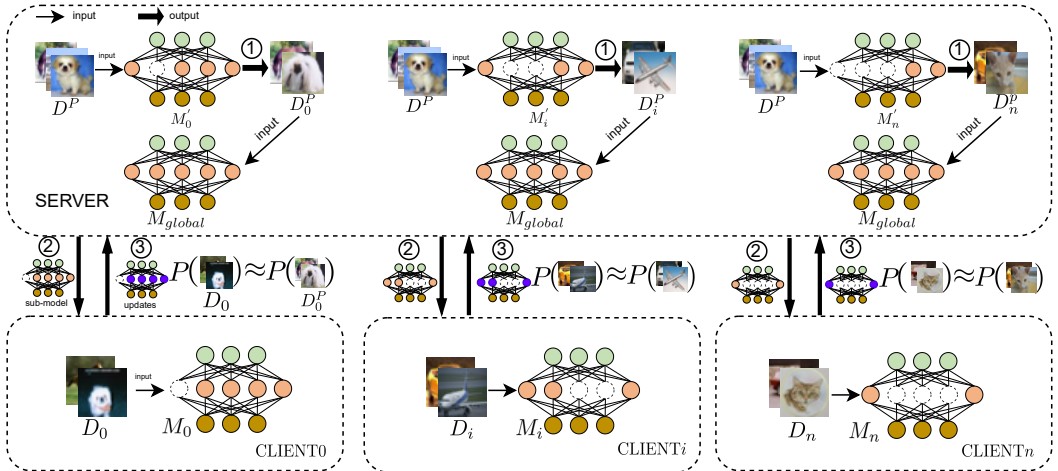

Figure 2: Overview of FedGSE framework. ① The server randomly selects communication client such as client $i$ and picks out $D_i^P$, which is a sub-dataset comes from $D^P$ and have a similar distribution with $D_i$, i.e. $P_{D_i^P} \approx P_{D_i}$. Server utilizes the most recently stored model $M_i^{'}$ to predict $D_i^P$. Sub-dataset can also be picked out by Client Send Local Labels to Server(CSL). ② servers employs global model $M_{global}$ to select neurons, these are then used to generated $M_i$ for client $i$ under the $D_i^p$. This process involves backward propagation and the selection of topk gradients layer by layer; ③ the clients train their models using their respective local datasets and send updates to the server. The server, in turn, updates the historic models $M_i^{'}$ and the global model $M_{global}$.

allocated based on specified rules, such as by order of neuron in layer. As a result, some neurons which unique to high-resource clients, may be trained significantly less frequently, which inevitably leads to a poorer performance of the model. For example, Fjord (Horvath et al., 2021) and Hetero employ a similar approach. The template of these two strategies is a neuron-order, which means that the neurons of each layer are allocated according to the capacity size from beginning to end. Clearly, the tail-end neurons only be trained in high-resource clients. Therefore, the uneven training of model neurons leads to a decrease in performance.

**Dynamic Allocation Strategy.** The dynamic allocation strategy differs from the static strategy in that it allocates neurons based on the current communication round rather than predetermined rules. But there have a unique method Federated Dropout, which prunes neurons randomly every communication round without considering the current situation. As mentioned earlier, the issue of uneven training of model neurons was observed with the Fjord and Hetero methods. The method FedRolex solve this problem using a rolling way for each neurons can be trained equally. It selectively chooses neurons for each layer in a rolling manner based on the number of communication rounds.

## 3 PRELIMINARIES

**Mathematical Representation.** We design the deep neural network have $L$ layers, and each layer have $I$ neurons. The weight parameters of the whole model are denoted by $\mathbf{w}$ and we denote the $\mathbf{w}_l$ as the weights of $i$-th, where $\mathbf{w}_l = [\mathbf{w}_{l,0}, \cdots, \mathbf{w}_{l,I-1}]$. And $\mathbf{w}_{l,i} = [w_{l,i}, b_{l,i}]$, $w_{l,i}$ and $b_{l,i}$ respectively represent weight and bias for $i$-th neuron of the $l$-th layer. Then its activate value output can be computed as $h_{l,i} = \sigma(w_{l,i}\mathbf{h}_{l-1} + b_{l,i})$, where $\sigma(\cdot)$ is the nonlinear activation functions (e.g., ReLU) and the $\mathbf{h}_{l-1}$ are the all activation output of the upper layer, i.e., $\mathbf{h}_{l-1} = [h_{l-1,0}, ..., h_{l-1,I-1}]$. It follows naturally, we user the $\mathbf{w}$ to represent all weights of the network (e.g., $\mathbf{w} = [\mathbf{w}_0, \dots, \mathbf{w}_{L-1}]$).

**Problem formulation.** We assume $N$ clients devices and each client have private non-IID(non-identically and independently distributed) local data $\mathbb{D} = \{\mathbb{D}_0, \mathbb{D}_1, \dots, \mathbb{D}_{N-1}\}$. The local dataset $\mathbb{D}_n = \{\mathbf{x_n}, \mathbf{y_n}\}$, where $\mathbf{x_n}$ and $\mathbf{y_n}$ represent the data and label of the local dataset. We use $x_n^i$ to denote $i$-th input data sample and $y_n^i \in \mathcal{C}\{0, 1, \dots, C\}$ to denote the corresponding label of $x_n^i$. Then $D_n = |\mathbf{x_n}|$, i.e., $D_n$ is the number of $n$-th client loacl dataset, and $D = \sum_{n=0}^{N-1} |\mathbf{x_n}|$. Finding

**w** to minimize the total empirical loss of the global model over the entire dataset $\mathbb{D}$ is our final goal:

$$\min_{w} F(\mathbf{w}) := \sum_{n=0}^{N-1} \frac{D_n}{D} F_n(\mathbf{w}), \ where \ F_n(\mathbf{w}) = \frac{1}{D_n} f(\mathbf{w}; \mathbf{x_n}, \mathbf{y_n})$$

where $f(\cdot)$ is the cross-entropy loss function that computes the difference between the predicted labels of $\mathbf{x_n}$ and ground-truth label $\mathbf{y_n}$, which is used to evaluate the fitting performance of the parameters **w**. It is obvious $F_n(\mathbf{w})$ denotes the average local loss function of $n$-th client.

## 4 FEDGSE ALGORITHM DESIGN

### 4.1 THE GRADIENT OF NEURON

In chapter3, we set $h_{l,i}$ to represent the $i$-th neuron of $l$-th layer. Specifically, $h_{l,i}$ is a feature map with two dimensions. The gradient of $h_{l,i}$ can be represented:

$$g_{l,i} = \sum_{k=0}^{K-1} \sum_{v=0}^{V-1} \left| \frac{\partial f(\mathbf{w}; \mathbf{x}, \mathbf{y})}{\partial h_{l,i}(k, v)} \right|$$

where $f(\mathbf{w}; \mathbf{x}, \mathbf{y})$ means the cross-entropy loss of deep neural network for the input $(\mathbf{x}, \mathbf{y})$. And the dimension length of feature map is $K$ and $V$, i.e., $h_{l,i} \in \mathbb{R}^{K \times V}$.

### 4.2 OBTAINING SIMILAR DATASET FROM PUBLIC DATASET

We construct public dataset(Zhu et al., 2021; Lin et al., 2020) in server side (See AppendixC for details). Public dataset encompasses all $\mathcal{C}$ classes. However, using the entire public dataset as similar data to calculate gradients has two drawbacks. Firstly, computing gradients for the entire public dataset is time-consuming. Secondly, there may not be a strong correlation between the public dataset and the non-iid data distribution of the client Based on the aforementioned reasons, we select the portion of data from the public dataset that exhibits the highest correlation with the client's data distribution as the similar data. We use $\mathbb{D}^P$ to represent public dataset, and $\mathbb{D}_i^P$ to represent the sub-dataset of the public dataset, which have the similar data distribution of client $i$ local dataset $\mathbb{D}_i$, then we obtain the formula:

$$P_{\mathbb{D}_i^P} \approx P_{\mathbb{D}_i} \tag{1}$$

We use two solutions to solve this problem.

**Client Send Local Labels to Server(CSL).** When server need to train a specific client, it sends a request to that client. The client then returns the most significant labels from its local dataset to server (Here we define $s$ as the number of significant labels). Finally, server select data that haves the same labels with client from public dataset as the similar data. Obviously, this method has disadvantages. Firstly, frequent communication leads to decreased efficiency. Secondly, there is a risk of privacy leakage when the client uploads the labels.

**Server Predicts Labels through historical Models(SPL).** This method is entirely executed on the server-side without requiring client participation. The server stores the latest trained sub-models from the clients. When the server needs to obtain client backward data, it loads the corresponding client sub-model. Then, it calculates the accuracy of each category of data in the public dataset using the client model. We consider the labels with top $S$ accuracy as the client local dataset labels to generate similar data. This method solve the disadvantages of method **CSL**.

### 4.3 PROGRESS OF SELECTING NEURONS

In server side, we utilize global model to get the neuron gradients **g** using the similar data (specifics in 4.2) from public dataset. How to get sub-model neurons from global model is the core of our method. Overall, the progress of our selecting is layer by layer like Figure2. In each layer, the gradient of neuron by the loss is the rule to select. The larger the gradient value of neuron we selected, the closer updating parameters are to global model. Furthermore, large gradient means

---

**Algorithm 1** `FedGSE`

---

**Input:** global model $\mathbf{w}$, learning rate $\eta$, total communication rounds $T$, all clients capacity $\beta = \{r_0, \cdots, r_{N-1}\}$, public dataset $\mathbb{D}^p = \{\mathbf{x^P}, \mathbf{y^P}\}$
**Output:** Trained global model $\mathbf{w_T}$

1: Initialize the model parameters $\mathbf{w_0}$
2: **procedure** SERVER-SIDE OPTIMIZATION
3:      **for** each communication round $t \in \{0, 1, \cdots, T-1\}$ **do**
4:          Randomly select a subset of clients $\mathcal{N}_t$
5:          **for** each selected client $n$ **in parallel do**
6:              $\mathbb{D}_n^p \leftarrow$ **GetSimilarDataCSL** $(\mathbb{D}^p, \mathbf{w^n})$ or $\mathbb{D}_n^p \leftarrow$ **GetSimilarDataSPL** $(\mathbb{D}^p)$     ▷
 In here $\mathbb{D}_n^p = \{ \mathbf{x_n^P}, \mathbf{y_n^P} \}$ . If **C**lients can **S**end local dataset **L**abels to server using function **GetSimilarDataCSL**3, else if **S**erver **P**redicts **L**abels using **GetSimilarDataSPL**2
7:              $\mathbf{M^{n,t}} \leftarrow$ **GetMask**4$(r_n, \mathbb{D}_n^p, \mathbf{w_t})$
8:              $\mathbf{w_t^n} \leftarrow \mathbf{w_t} \odot \mathbf{M^{n,t}}$
9:              $\mathbf{w_{t+1}^n} \leftarrow$ **ClientLocalUpdata**$(n, \mathbf{w_t^n})$
10:          **end for**
11:          updata the global model $\mathbf{w_{t+1}} = \sum_{n \in \mathcal{N}_t} \mathbf{P_t^n} \odot \mathbf{w_{t+1}^n}$
12:      **end for**
13: **end procedure**
14: **procedure** CLIENTLOCALUPDATE$(n, \mathbf{w_t^n})$
15:      Receive $\mathbf{w_t^n}$ from the server
16:      **for** each local iterations $e$ from 1 to $E$ **do**
17:          Updata sub-model parameters on private data $\mathbf{w_{t,e+1}^n} = \mathbf{w_{t,e}^n} - \eta \nabla_{\mathbf{w_{t,e}^n}} f_n(\mathbf{w_{t,e}^n})$
18:      **end for**
19:      **return** $\mathbf{w_{t,E+1}^n}$
20: **end procedure**

---

that changing these neurons have a greater impact on the loss compared to other neurons(Selvaraju et al., 2017). We set client model capacity ratio $r(0 < r \leq 1)$, which means the maximum ratio of clients that can train a global model. More precisely, given a layer $l$ and a client $n$, we only retain the top ratio $r$ of neurons (i.e., $\max_{i=1}^{r \cdot I} g_{l,0}, g_{l,1}, \ldots, g_{l,I-1}$) with the highest gradient values $g_{l,i}$, while pruning the other neurons (show in Figure2). This operation results in a sub-model $\mathbf{w^n}$, which is obtained by element-wise multiplying the original weights $\mathbf{w}$ with a mask $\mathbf{M^n}$, where $\mathbf{M^n}$ (generate by the algorithm4) is the binary mask corresponding to the different neurons. If the $l$-th layer $i$-th neuron is pruned, $\mathbf{M_{l,i}^n} = 0$, and $\mathbf{M_{l,i}^n} = 1$ otherwise. For single neuron, $\mathbf{w_{l,i}^n} = \mathbf{w_{l,i}} \odot \mathbf{M_{l,i}^n}$. For sub-model, $\mathbf{w^n} = \mathbf{w} \odot \mathbf{M}$.

### 4.4 TRAIN AND AGGREGATE

Clients receives the parameters of sub-models from server. Then use whole local dataset to train model locally. The client updates the sub-model $\mathbf{w^n} = \mathbf{w^n} - \eta \nabla_{\mathbf{w^n}} f_n(\mathbf{w^n})$, where $\mathbf{w^n}$ is the parameters of sub-model, $\eta$ is the learning rate and $f_n(\cdot)$ is the loss. Finally, the clients send their updated parameters of sub-model back to server. Server aggregates these parameters to update the global model: $\mathbf{w} = \sum_{n \in \mathcal{N}} \mathbf{P_t^n} \odot \mathbf{w^n}$, where $\mathcal{N}$ is a subset of clients which participate training sub-model, and $\mathbf{p_{l,i}^n} = \dfrac{D_n}{\sum_{i \in \mathcal{N}} D_i}$. i.e. The aggregation coefficient of client $n$, which represents the proportion of the number of client $n$ data in clients $\mathcal{N}$.

## 5 THEORETICAL ANALYSIS

**Lemma 1** *The gradients of the parameters are proportional to the gradients of their corresponding activation values.* $\mathbf{k}$ *is a proportionality coefficient.*

$$\frac{\partial f}{\partial w_{l,i}} = \mathbf{k} \cdot g_{l,i}, \qquad (2)$$

The proofs are deferred to Appendix B. Lemma 1 indicates that there is a linear relationship between $\frac{\partial f}{\partial w_{l,i}}$ and $g_{l,i}$. This means that choosing a relatively large $g_{l,i}$ is equivalent to choose a relatively large $\frac{\partial f}{\partial w_{l,i}}$.

**Theorem 1** *According to Algorithm4, we can get the sub-model parameters $\mathbf{w}_*^n$. Base Lemma 1, we can manifestly acquire that under $\mathbb{D}_n^p$, the gradient of $\mathbf{w}_*^n$ is closest to the global parameters $\mathbf{w}$. Because equation1, we get:*

$$\|\nabla f(\mathbf{w}; \mathbb{D}_n) - \nabla f(\mathbf{w}^n; \mathbb{D}_n)\| \geq \|\nabla f(\mathbf{w}; \mathbb{D}_n) - \nabla f(\mathbf{w}_*^n; \mathbb{D}_n)\|, \tag{3}$$

*where $\mathbf{w}^n$ is the parameters of sub-model, and $\mathbb{D}_n$ is the client $n$'s local dataset.*

The specific proof is placed in the AppendixB. Theorem 1 proves that using the sub-model selected by our algorithm 1 to update model on local dataset is closest to directly update model using origin global model to train the local dataset.

## 6 EXPERIMENTS

**Datasets and Model.** In this study, we assess the effectiveness of the proposed `FedGSE` by conducting experiments on two different models and three widely-used datasets. The two distinct models is CNN to train EMNIST (LeCun, 1998) and pre-activated ResNet18 Shafiq & Gu (2022) to train CIFAR-10 and CIFAR-100 Krizhevsky et al. (2009). We use the Static Batch Normalization method instead of Batch Normalization Diao et al. (2020); Andreux et al. (2020) and introduce a Scalar module after each convolution layer. Specially, the CNN model is simply composed by four convolution layers, which channels are $\{64, 128, 256, 512\}$ respectively.

**Client Local Data Distribution.** We adopted the approach proposed by HeteroFL to represent non-IID data distributions Li et al. (2020); Diao et al. (2020)in all datasets, which limited each client to $s$ labels. According to the value of $s$, we define High Data Heterogeneity and Low Data Heterogeneity. The smaller the value of $s$, the greater the data heterogeneity. The special definitions show in Table 1. In the following experiments, we will also compare the impact of different other $s$ values on the overall global accuracy.

Table 1: Corresponding table of $s$ and data heterogeneity.

| Dataset | EMNIST | CIFAR-10 | CIFAR-100 |
|---|---|---|---|
| High Data Heterogeneity | 2 | 2 | 8 |
| Low Data Heterogeneity | 4 | 4 | 16 |

**Model heterogeneity.** Without loss of generality, we set five different client capacities $\beta_n \in \{1, 1/2, 1/4, 1/8, 1/16\}$, which means clients can obtain how much of the global model. To produce client models, we experiment by adjusting the number of kernels in the convolution layers while maintaining a consistent number of nodes in the output layers.

**Baselines.** We compare three predetermined rule methods, HeteroFL, Federated Dropout and FedRolex. To ensure a fair comparison, we adopt identical parameters for all experiments, including the learning rate, local epochs, and number of communication rounds. Especially, the number of similar data in `FedGSE` is far less than the number of client local dataset. In order to focus on the method itself, we used a constant learning rate and did not use the existing multi-step learning rate decay schedule. Additional information regarding each method and dataset can be found in AppendixE.

**Configurations and platform.** To augment the images in EMNIST, CIFAR-10, and CIFAR-100, we apply bounding box crop Lambeta et al. (2021). During each communication round, $10\%$ of the 100 clients are randomly chosen for training, with a fraction (frc) of $10\%$. At the start of each communication round, the selected clients' capabilities are dynamically selected from a uniform distribution. The experiments were conducted using the PyTorch framework(Paszke et al., 2019), and the specifications of the hyperparameters can be found in the AppendixE. The computations were performed on machines equipped with Nvidia RTX 4090 and K80 GPUs.

**Evaluation metric.** Image classification tasks were evaluated using global accuracy, which is defined as the accuracy of the server model over the entire test set.

## 6.1 Performance Comparison with State-of-the-Art Methods.

Table 2: The comparison of global test accuracy on different datasets. `FedGSE` public dataset come from clients local dataset using **Clients Upload partial Data** in AppendixC. `FedGSE` (1) represent **CSL** in 4.2, `FedGSE` (2) represent **SPL** in 4.2. The number of similar data in `FedGSE` is 128.

| Method | High Data Heterogeneity(%) | | | Low Data Heterogeneity(%) | | |
|---|---|---|---|---|---|---|
| | **EMNIST** | **CIFAR-10** | **CIFAR-100** | **EMNIST** | **CIFAR-10** | **CIFAR-100** |
| HeteroFL | 96.11 | 56.08 | 25.95 | 98.65 | 73.42 | 32.13 |
| Federated Dropout | 88.76 | 52.57 | 15.79 | 97.53 | 65.28 | 19.81 |
| FedRolex | 93.94 | 57.28 | 21.53 | 98.56 | 71.68 | 27.60 |
| FedGSE (1) | **98.06** | **65.89** | **31.46** | 98.74 | 73.51 | **32.65** |
| FedGSE (2) | 97.44 | 65.82 | 28.87 | **98.77** | **74.66** | 32.40 |

The table 2 show the result of our experiments those are executed in the same setting to ensure fairness. And the results demonstrate that our method `FedGSE` performs better than others, especially in high data heterogeneity region, which exceeds performance by 1.33%, 8.54% and 2.92% comparing to the best performance of Hetero, Federated Dropout and FedRolex corresponding to EMNIST, CIFAR10 and CIFAR100. Beside, our method results also outperforms in low data heterogeneity region, have 0.12%, 1.24% and 0.27% advantage than best of the other three method in EMNIST, CIFAR10 and CIFAR100. The above phenomena indicate our method have a good advantage in high data heterogeneity region, which validated our theory that the more heterogeneous the data, the better our method could select neurons important for modeling those data. As we can see, the Federated Drop generally performers relatively worse in most scenarios than others. The dynamic selecting neurons rule account for such result. For the simple dataset EMNIST, the accuracy's difference of our method and other method are small whether the high or low data heterogeneity, that because it is easy for CNN model to train EMNIST so that all method can get relatively better results. However in middle dataset CIFAR10, `FedGSE` demonstrated its superiority in algorithm design, and have 8.54% and 2.98% than FedRolex. As the heterogeneity of data decreased, the advantage narrowed, but still maintained its leading position. Although on difficult dataset CIFAR-100, the performances of all methods were not as good as on other datasets, our method still remained ahead of other methods. Overall, `FedGSE` consistently outperforms HeteroFL, Federated Dropout and FedRolex under both low and more challenging high data heterogeneity scenarios, especially under high. The above analysis is based on `FedGSE` (2). The comparing of `FedGSE` (1) and `FedGSE` (2) will be introduced in next section.

## 6.2 The comparison of FedGSE (1) and FedGSE (2)

Under high data heterogeneity, `FedGSE` (2), which server predicts labels through previous models, could not achieve the level of `FedGSE` (1), which clients send labels to server. In high data heterogeneity, client dataset contains a small number of classes(e.g., s is 2 in EMNIST and CIFAR10, and 5 in CIFAR100), so it is difficult for server to predict client data distribution fully accurately. That incur above result. But in low data heterogeneity, client dataset contains more number of classes showed in table1. This provided the a method with robustness tolerance, where a small number of prediction errors did not impact the overall neuron selection. The evaluation results show `FedGSE` (1) and `FedGSE` (2) achieve the same level.

## 6.3 Impact of Client Model Heterogeneity Distribution.

In previous experiments, clients capacities are set uniformly. To figure out the influence of the client model heterogeneity distribution, we choose the capacities $\beta = \{1, 1/16\}$ and range the distribution ratio between the two (defined as $\rho$), where $\rho = 1$ means represents the case in which all the clients have the largest capacity model ($\beta = \{1\}$) and $\rho = 0$ means the case in which all the clients have the smallest capacity model ($\beta = \{1/16\}$).

**High and low data heterogeneity comparison under `FedGSE`.** Figure3(a)3(d) show the global model changes when $\rho$ varies from 0 to 1 in EMNIST and CIFAR10 using our method. We can observe that (1) as $\rho$ varies from 0 to 0.2, global model accuracy experience a notable leap whether high or low data heterogeneity in both EMNIST and CIFAR10. That indicate model scale is the most reason incurring above situation. (2) for simple dataset EMNIST, there is a gap in global accuracy between high and low data heterogeneity for a wide range of $\rho$ (from 0 to 1). This is because EMNIST is a simple task and CNN model can get 85% global model accuracy easily. Hence the global model accuracy is bottlenecked by the level of data heterogeneity instead of model capacity. This results reveal the challenges of data heterogeneity can not be addressed by increasing the capacities of model. (3) for CIFAR10, there have a large gap in global accuracy between high and low data heterogeneity. The global model accuracy is bottlenecked by the highest capacity of the models and the level of data heterogeneity. This lead to the large gap.

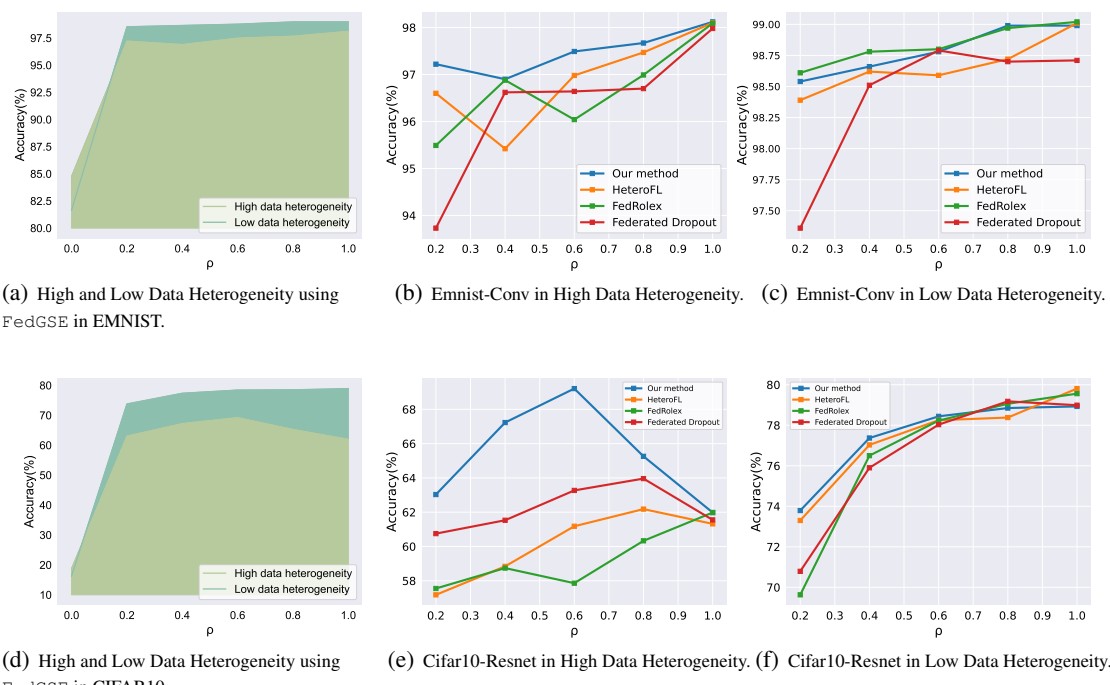

(a) High and Low Data Heterogeneity using `FedGSE` in EMNIST.

(b) Emnist-Conv in High Data Heterogeneity.

(c) Emnist-Conv in Low Data Heterogeneity.

(d) High and Low Data Heterogeneity using `FedGSE` in CIFAR10.

(e) Cifar10-Resnet in High Data Heterogeneity. (f) Cifar10-Resnet in Low Data Heterogeneity.

Figure 3: Comparison of client model capacity heterogeneity distribution in EMNIST and CIFAR10 under different methods.

**Comparison of different methods.** As show in figure3(b)3(e), in the scenario of high data heterogeneity, our method achieves overwhelming advantages compared with $\rho$ throughout the process of changing from 0.2 to 1. This shows that increasing the model's capabilities does not necessarily improve global accuracy, and in high scenarios, good algorithm design is truly needed to improve the model's accuracy. In difficult scenarios, it far exceeded other models, proving that our algorithm has more advantages in such scenarios. While in the scenario of low data heterogeneity as showed by figure3(c)3(f), our method achieves relatively good performance in most cases. In this kind of simple scenario, most methods achieved good results, which shows that in this current scenario, limitations on global accuracy come from the model's capabilities, and improving the model's capabilities can improve the effectiveness. Especially in Figure3(e), in difficult datasets with high data heterogeneity, increasing the model's capabilities cannot stably improve the model's accuracy. In summary, in stringent scenarios targeting high data heterogeneity, our method would be more applicable than other methods.

## 6.4 IMPACT OF CLIENT DATASET DISTRIBUTION.

To investigate the impact of the degree of data heterogeneity on model accuracy, we do the Table3 experiments. We vary $s$ (the number of containing classes in client local dataset) from 2 to 10. From the experimental results, we can see that when $s$ varied from 2 to 4, the global model accuracy undergo a dramatic jump, after which the accuracy remain in a stable stage. The experimental results indicate that there exists a threshold in terms of the degree of data heterogeneity's impact on the model, when this threshold is exceeded, the level of influence decreases. Clearly, the threshold in EMNIST and CIFAR10 are all $s = 2$.

Table 3: Impact of client dataset distribution in EMNIST and CIFAR10.

| Dataset | The containing classes of client dataset | | | | |
|---|---|---|---|---|---|
| | 2 | 4 | 6 | 8 | 10 |
| emnist-conv | 97.44 | 98.77 | 98.94 | 98.81 | 99.01 |
| cifar10-resnet | 65.82 | 74.66 | 75.04 | 76.38 | 74.38 |

## 6.5 IMPACT OF THE NUMBER OF COMMUNICATION CLIENTS.

To explore the impact of the number of selecting communication clients per round in training process, we set $frc$ (means the proportion of communication clients against total clients) in $\{0.05, 0.10, 0.15, 0.20\}$.

Table4 present our result. There have two conclusions. (1) When $frc = 0.05$, the global model accuracy perform much lower than $frc = \{0.10, 0.15, 0.20\}$. This indicates that when the number of communication clients is relatively small, it leads to fewer parameters being updated in global model, thereby decreasing the accuracy. (2) When increasing $frc$ from $0.10$, the global model accuracy remain a relatively same level, sometime, the accuracy decreases. That because when $frc$ exceed $0.1$, there have enough communication clients parameters being updated in global model, so the accuracy remain in same level. When increasing $frc$ continuously, the competition of parameters increase, sometimes incur accuracy decreases.

Table 4: Impact of communication clients in EMNIST and CIFAR10.

| Dataset | Data Heterogeneity | The proportion of communication clients per rounds | | | |
|---|---|---|---|---|---|
| | | 0.05 | 0.10 | 0.15 | 0.20 |
| emnist-conv | High | 96.92 | 97.44 | 97.66 | 97.49 |
| | Low | 98.55 | 98.77 | 98.84 | 98.80 |
| cifar10-resnet | High | 60.87 | 65.82 | 63.36 | 63.19 |
| | Low | 69.34 | 74.66 | 75.80 | 76.12 |

## 7 CONCLUSION AND DISCUSSIONS

In this paper, we focus on the updates relationship of global and sub-model. Then we design the algorithm `FedGSE` which aims to make the sub-model updates approximate to the global. And we theoretically proved that the parameters updated by our method are closest to the global model. To validate this idea, we use CNN and Resnet18 to conduct experiments under EMNIST, CIFAR10 and CIFAR100. Extensive experiments results verify that our method performs optimally in multifarious extraction neurons sub-model methods. Additionally, we conduct further experiments to compare the importance of different Hyperparameters such as, model heterogeneity, client data heterogeneity, similar data scale and number of communication clients on our method.

Nevertheless, there are still some limitations that need further improvement in the future. For instance, our method requires the construction of a public dataset on the server side, which may be challenging for domains with limited data availability. Furthermore, executing the backward progress on the server side necessitates matching capabilities, which raises the application threshold. In future work, we will make efforts to address and optimize these issues.

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

## A   GET BACKWAD DATA ALGORITHM

---

**Algorithm 2** GetSimilarDataSPL

---

**Input:** public dataset $\mathbb{D}^p = \{\mathbf{x}^P, \mathbf{y}^P\}$, client model $\mathbf{w^n}$
**Output:** client n backward data $\mathbb{D}_n^p$
1: **procedure** GETSIMILARDATASPL($\mathbb{D}^p, \mathbf{w^n}$)
2:     **if** $\mathbf{w^n}$ not Initialization **then**
3:         Randomly select $s$ classes to compose subset $\mathcal{L}$ from $\mathcal{C}$
4:     **else**
5:         **for** each class $c \in \mathcal{C} = \{0, 1, \cdots, C\}$ **do**
6:             $\hat{\mathbf{y}}_c = f_n(\mathbf{w^n}; \mathbf{x}^{P,\{\mathbf{c}\}})$
7:             $\mathcal{A} \leftarrow \mathcal{A} \cup \{\frac{|\hat{\mathbf{y}}_c = \mathbf{y}^{P,\{\mathbf{c}\}}|}{|\mathbf{y}^{P,\{\mathbf{c}\}}|}\}$          ▷ The accuracy of class $C$ on public dataset.
8:             get the label indexes $\mathcal{L}$ according to $\overset{s}{\max} \mathcal{A}$     ▷ set $\mathcal{L}$ contains the top $s$ element's index of $\mathcal{A}$
9:         **end for**
10:    **end if**
11:    **return** $\mathbb{D}_n^p \leftarrow \{\mathbf{x}^{P,\mathcal{L}}, \mathbf{y}^{P,\mathcal{L}}\}$                    ▷ In here $\mathbf{x}^{P,\mathcal{L}} = \mathbf{x_n^P}, \mathbf{y}^{P,\mathcal{L}} = \mathbf{y_n^P}$
12: **end procedure**

---

**Algorithm 3** GetSimilarDataCSL

---

**Input:** public dataset $\mathbb{D}^p = \{\mathbf{x^P}, \mathbf{y^P}\}$
**Output:** client n backward data $\mathbb{D}_n^p$
1: **procedure** GETSIMILARDATACSL($\mathbb{D}^p$)
2:     $\mathcal{L} \leftarrow$ **ClientSendLabel5**
3:     **return** $\mathbb{D}_n^p \leftarrow \{\mathbf{x}^{P,\mathcal{L}}, \mathbf{y}^{P,\mathcal{L}}\}$                    ▷ In here $\mathbf{x}^{P,\mathcal{L}} = \mathbf{x_n^P}, \mathbf{y}^{P,\mathcal{L}} = \mathbf{y_n^P}$
4: **end procedure**

---

**Algorithm 4** GetMask

---

**Input:** client $n$ capacity $r_n$, public dataset $\mathbb{D}^p = \{\mathbf{x^P}, \mathbf{y^P}\}$, $t$ round global model $\mathbf{w_t}$
**Output:** client $n$ mask $\mathbf{M^{n,t}}$ in $t$-th round
1: **procedure** GETMASK($r_n, \mathbb{D}_n^p, \mathbf{w_t}$)
2:     $\mathbf{g^n} = \nabla_{\mathbf{h}} f(\mathbf{w_t}; \mathbf{x_n^P}, \mathbf{y_n^P})$
3:     **for** each layer $l \in \{0, 1, \cdots, L-1\}$ **do**
4:         $\mathcal{M} \leftarrow \overset{r_n \cdot I_l}{\underset{i=0}{\max}} g_{l,0}^n, g_{l,1}^n, \ldots, g_{l,I_l-1}^n$
5:         **if** $\mathbf{g_{l,i}^n} \in \mathcal{M}$ **then**
6:             $\mathbf{M_{l,i}^{n,t}} = 1$
7:         **else**
8:             $\mathbf{M_{l,i}^{n,t}} = 0$
9:         **end if**
10:    **end for**
11:    **return** $\mathbf{M^{n,t}}$
12: **end procedure**

---

## B   THEOREM PROVE

$$h_{l,i} = \max\{0, w_{l,i}^T \mathbf{h}_{l-1} + b_{l,i}\} \tag{4}$$

$$\frac{\partial f}{\partial w_{l,i}} = \frac{\partial f}{\partial h_{l,i}} \frac{\partial h_{l,i}}{\partial w_{l,i}} = \frac{\partial f}{\partial h_{l,i}} \mathbf{h}_{l-1} = g_{l,i} \mathbf{h}_{l-1} \tag{5}$$

---

**Algorithm 5** ClientSendLabel

---

1: **procedure** CLIENTSENDLABEL
2:      **return** local dataset labels $\mathcal{L}$
3: **end procedure**

---

Equation4 represent the full connection with ReLU activation. And the equation5 the SGD backward process, there is a linear relationship between $\frac{\partial f}{\partial w_{l,i}}$ and $g_{l,i}$. This means that choosing a relatively large $g_{l,i}$ is equivalent to choosing a relatively large $\frac{\partial f}{\partial w_{l,i}}$.

$$\mathbf{M} = \mathbf{GetMask4}(r, \mathbb{D}_n^p, \mathbf{w}) \tag{6}$$

$$\mathbf{w}_*^n = \mathbf{M} \cdot \mathbf{w} \tag{7}$$

Equation6 get the mask $\mathbf{M}$ of sub-model using our design strategy. Then we use equation7 to get the sub-model parameters $\mathbf{w}_*^n$. Evidently, the equation8 is valid.

$$\|\nabla f(\mathbf{w}; \mathbb{D}_n^p) - \nabla f(\mathbf{w}^n; \mathbb{D}_n^p)\| \geq \|\nabla f(\mathbf{w}; \mathbb{D}_n^p) - \nabla f(\mathbf{w}_*^n; \mathbb{D}_n^p)\| \tag{8}$$

According to the equation1, we finally prove the equation9:

$$\|\nabla f(\mathbf{w}; \mathbb{D}_n) - \nabla f(\mathbf{w}^n; \mathbb{D}_n)\| \geq \|\nabla f(\mathbf{w}; \mathbb{D}_n) - \nabla f(\mathbf{w}_*^n; \mathbb{D}_n)\| \tag{9}$$

## C    PUBLIC DATASET

In 4.1, we define the gradient of neuron. As we all know, the gradients of neurons come from the **backward** progress in training model section. Furthermore, it pertains to the **global model** of backward. Backward necessitates greater computational capacity and memory compared to forward. So executing backward progress in client become an important thing. To address this problem, we put backward process on server. Then there have a question. Federated Learning is a paradigm that protect the clients private information. This means the server have no data to get gradients. Therefore, we should establish a Public Dataset on the server side. There are three alternative approaches to create Public Dataset:

**Clients Upload partial Data**. The direct method is that, clients upload partial local dataset to server. Uploading a fraction of the data amounting to one percent of the local data is deemed sufficient. This data should have label.

**Generating Counterfeit Data.** client can using local data to generate counterfeit data base generative model, such as DDPM(Ho et al., 2020), GAN(Creswell et al., 2018), Stable Diffusion-croitoru2023diffusion and so on. This generative model can learn the local dataset distribution and then output the similar data. This approach not only prevents privacy breaches but also yields data that closely resembles the original. But some generative model is to big for some client to play it.

**Employing Analogous Data.** We can opt for alternative datasets that encompass similar data types or incorporate local data labels as a substitute. For example, if the local dataset is CIFAR10, then ImageNet(Deng et al., 2014) can serve as a viable alternative.

## D    IMPACT OF THE SIMILAR DATA SCALE.

We set different number of the similar data to address how this affect the global model accuracy. The result Table5 show that when the number of the similar data vary from $64$ to $512$, the global model accuracy do not have a notable improvement. This means that we could appropriately reduce the amount of the public dataset, and that can not significantly degrades the global model accuracy, while also improving computational speed and reduce computational resource requirements.

## E    MORE EXPERIMENTAL DETAILS

All figures and tables related to experiments in the paper adopt the setting of table6.

Table 5: Impact of similar data scale in EMNIST and CIFAR10.

| Dataset | Data Heterogeneity | The number of similar data | | | |
| --- | --- | --- | --- | --- | --- |
| | | 64 | 128 | 256 | 512 |
| emnist-conv | High | 96.61 | 97.44 | 96.55 | 96.39 |
| | Low | 98.75 | 98.77 | 98.82 | 98.83 |
| cifar10-resnet | High | 64.55 | 65.82 | 63.36 | 63.19 |
| | Low | 74.44 | 74.66 | 74.42 | 74.65 |

Table 6: Experimental setup details on EMNIST, CIFAR-10 and CIFAR-100.

| | | EMNIST | CIFAR-10 | CIFAR-100 |
| --- | --- | --- | --- | --- |
| Local Epoch | | 2 | 2 | 2 |
| Batch Size | | 16 | 16 | 16 |
| Learning Rate | | 0.001 | 0.001 | 0.001 |
| Decay Schedule | High Data Heterogeneity | None | None | None |
| | Low Data Heterogeneity | None | None | None |
| Communication Rounds | High Data Heterogeneity | 800 | 2500 | 2500 |
| | Low Data Heterogeneity | 800 | 2500 | 2500 |
| Optimizer | | SGD | SGD | SGD |
| Momentum | | 0.9 | 0.9 | 0.9 |
| Weight Decay | | 5.00E-04 | 5.00E-04 | 5.00E-04 |
| Similar data | | all | all | all |

The figure4 show the sum of global and sub-model parameters, which do experiments in EMNIST using CNN. As we can see, our method have a low sum of parameters both in global model and sub-model, which indicates our method optimizes parameters more effectively than other methods.

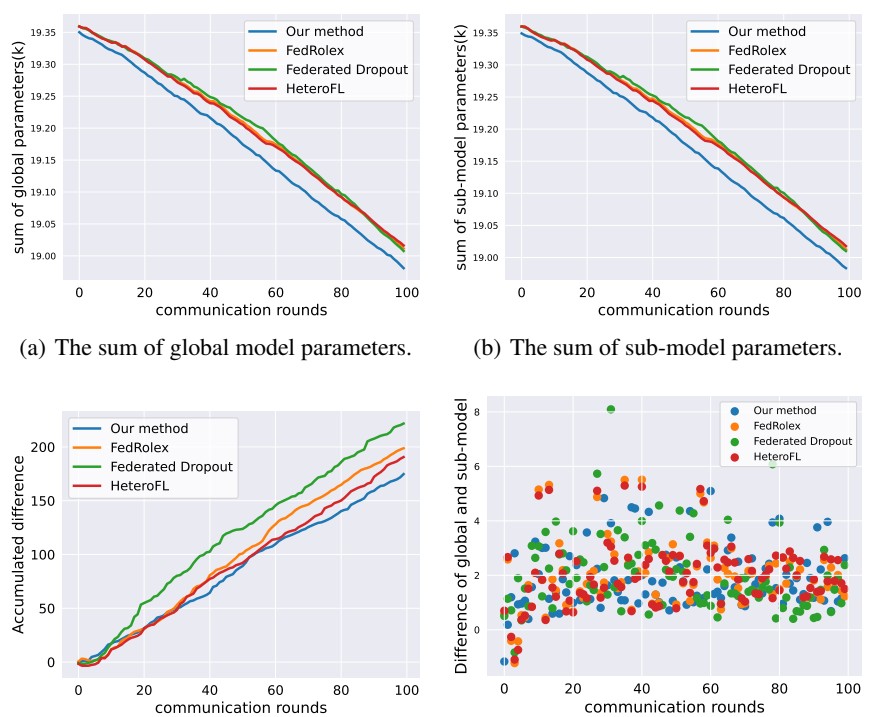

(a) The sum of global model parameters.

(b) The sum of sub-model parameters.

(c) The cumulative errors of parameter difference from every communication rounds.

(d) The scatter plot of parameter difference.

Figure 4: Supplementary data of Figure1.

