# OpenReview forum: "FedGSE:Gradient-based Sub-model Extraction for Resource-constrained Federated Learning"
_ICLR.cc/2024/Conference — ICLR 2024 Conference Withdrawn Submission_

### Official Review · Reviewer_wpSY · 2023-10-23

**Soundness:** 2 fair
**Presentation:** 2 fair
**Contribution:** 2 fair
**Rating:** 5
**Confidence:** 4

**Summary:**

This work proposes a gradient-based federated sub-model extration to save computation cost of pervasive client devices. The motivation holds which helps the deployment of federated learning in the edge computation scenarios.

**Strengths:**

- The motivation to save computation cost of federated learning holds especially in the edge computing scenarios.
- Authors provide the implementation codes of their proposed method.

**Weaknesses:**

- There are numerous typos in the writing, and the authors should check more before they submit their work. Meanwhile, authors are encouraged to improve the writing and organization.
- The related work dicussion is poor. Authors are recommended to read a recently published paper [1] to learn the related organization of related works. Authors are encouraged to learn how to cite the related papers correctly.
- The assumption of this work is too ideal. Firstly, the acquisition of a benchmark dataset at the server side is hard and usually viewed as an ideal assumption. Secondly, the label distribution of each client is also a a "secret" for each client, which may be not shared to the server due to privacy consideration.
- The evaluation details are not clear and correct enough. Firstly, all experiments should be implemented on the same type of GPUs to ensure a fair comparison. Secondly, the Non-iid setting is relatively poor. Authors should first state clearly that data are partitioned via Dirichlet distribution. Authors should follow the mainstream seletion for s (e.g. 0.1-1.0).

[1] Xue J, Liu M, Sun S, et al. FedBIAD: Communication-Efficient and Accuracy-Guaranteed Federated Learning with Bayesian Inference-Based Adaptive Dropout[C]//2023 IEEE International Parallel and Distributed Processing Symposium (IPDPS). IEEE, 2023: 489-500.

**Questions:**

See above.

---

### Official Review · Reviewer_y9Vb · 2023-10-30

**Soundness:** 2 fair
**Presentation:** 3 good
**Contribution:** 2 fair
**Rating:** 3
**Confidence:** 4

**Summary:**

The paper presents FedGSE, a novel method for gradient-based sub-model extraction in federated learning, aiming to address the challenge of resource-constrained clients. The authors propose a strategy that selects neurons within each layer of the global model based on their large gradients generated by training the global model on a public dataset. The selected neurons are then used to form sub-models for training on the client side using local datasets. This ensures that the gradient updates produced by the sub-model closely resemble the gradient updates that would be produced when training the client data on the global model, resulting in better alignment between the sub-model and global model performance.

The key contributions of the paper are:
- Proposing the *first* method to improve federated learning training by approximating the update gradients between the global model and sub-models.
- Mathematically proving that the designed algorithm can make the update gradients between the global model and sub-models closest.
Validating the efficiency of the proposed method through extensive experiments, demonstrating that FedGSE outperforms other state-of-the-art methods.
- Validating the efficiency of the proposed method through extensive experiments, demonstrating that FedGSE outperforms other state-of-the-art methods.

**Strengths:**

**Originality:**
The paper introduces FedGSE, a novel method for gradient-based sub-model extraction in federated learning. The authors focus on optimizing the update gradients between the global model and sub-models, which is a unique approach compared to existing methods that primarily focus on designing allocation strategies for training workload. The proposed method demonstrates originality in its problem formulation and solution.

**Quality:**
The paper is well-structured and presents a clear and coherent explanation of the proposed method. The authors provide a comprehensive review of related works, highlighting the limitations of existing methods and how FedGSE addresses them. The experimental results demonstrate the effectiveness of the proposed method, and the authors provide a thorough analysis of the results, including the impact of various hyperparameters on the performance of the method.

**Clarity:**
The paper is written in a clear and concise manner, making it easy for readers to understand the proposed method. The authors provide a detailed description of the algorithm and its components, as well as the experimental setup and results. The figures and tables are well-designed and partially support the main points of the paper.

**Significance:**
The proposed FedGSE method has the potential to advance the field of federated learning, particularly in addressing the challenge of resource-constrained clients. By optimizing the update gradients between the global model and sub-models, the method demonstrates improved performance over existing methods. The experimental results show that FedGSE consistently outperforms other state-of-the-art methods, especially in high data heterogeneity scenarios. This indicates that the proposed method has significant potential for real-world applications and can contribute to the advancement of federated learning research.

**Weaknesses:**

### General

**Claim of Novelty:** The authors assert that their method is pioneering in enhancing Federated Learning training by approximating the update gradients between the global model and sub-models. This claim hinges on the novelty of their strategy, which zeroes in on the optimization direction of approaching the global model's effectiveness when extracting sub-models. While they have provided a thorough review of related works, including static and dynamic allocation strategies for sub-model extraction, a significant oversight is the lack of comparison with other resource-constrained federated learning methods, especially those based on Knowledge Distillation (KD).It would be beneficial for the authors to expand their discussion and benchmarking to include KD-based methods and other strategies that address resource constraints in federated learning.

** Scalability Concerns:** The paper addresses a highly practical scenario of resource-constrained FL. However, the datasets used for validation, such as CIFAR-10/100, may not be sufficiently large or complex to truly test the scalability of the proposed method. For a method targeting resource-constrained environments, it's crucial to demonstrate its efficacy on high-dimensional data, such as face recognition datasets like CelebA or ImageNet.Suggestion: The authors should consider conducting experiments on larger and more complex datasets to truly validate the scalability and robustness of their approach.

**Potential Limitations**: The authors have acknowledged some limitations of their method, such as the need for constructing a public dataset on the server side and the challenges associated with executing the backward progress on the server side. While it's commendable that they've recognized these issues, it's essential to delve deeper into potential solutions or workarounds for these challenges.

### About Algorithm

**Gradient Representation and Assumptions:** The paper introduces the gradient of a neuron with the equation
$g_{l,i} = \sum_{k=0}^{K-1} \sum_{v=0}^{V-1} |\frac{\partial f(w; x, y)}{\partial h_{l,i}(k, v)}|$
where $f(w; x, y)$ represents the cross-entropy loss of the deep neural network for the input $(x, y)$. The dimension length of the feature map is represented by $K$ and $V$, i.e., $h_{l,i} \in R^{K×V}$. However, the paper does not provide a clear justification or empirical evidence for choosing this gradient representation. It would be beneficial to have a more detailed discussion.

**Proof**: The proofs of Lemma 1 and Theorem 1 share Appendix B, but it is oversimplified, and not clear to see its pertinence. Moreover, how can eq (5) be derived from eq (4) with a non-linear relationship?

**Questions:**

1. The authors claim that their method is the first to propose improving Federated Learning training by approximating the update gradients between the global model and sub-models. This claim is based on the novelty of their approach, which focuses on designing a strategy that starts from the optimization direction of approaching the effectiveness of the global model when extracting sub-models. The authors provide a comprehensive review of related works, including static and dynamic allocation strategies for sub-model extraction, and highlight the limitations of these methods. In my opinion, the compared benchmark should also include the other resource-constrained (or heterogeneous) federated learning methods (like KD-based) apart from sub-module extraction methods. Could the authors elaborate more discussion on these methods?
2. (*miscellaneous*) In the text of this paper, the brackets are directly adjacent to the word without any space, which is unconventional. The numbers following the Table/Figure should also be separated by a space. The reference formatting is chaotic (e.g., the algorithm references in Algorithm 1).
3. The paper works on a very practical scenario of resource-constrained FL. In this case, it is of high importance to verify its scalability to high-dimensional data wit (e.g., face recognition datasets like CelebA or ImageNet). Datasets like CIFAR-10/100 are too small to verify its scalability.
4. The paper should provide information about the extra local training cost when applying GetSimilarDataCSL/SPL, as **resource-constrained** clients, not only the uploaded parameters but also local computational cost should also be considered, unless the setting of this paper is "communication-constrained" FL.

---

### Official Review · Reviewer_tC96 · 2023-11-01

**Soundness:** 3 good
**Presentation:** 3 good
**Contribution:** 3 good
**Rating:** 6
**Confidence:** 3

**Summary:**

The authors propose FedGSE a method that is able to extract sub-models from the entire model to improve performance. The method claims to be the first that exploits the observation that the update gradients between global and sub-models can be used to improve the FL scheme.

**Strengths:**

The work is timely and the topic is rather interesting. I personally followed closely the works that stemmed from Hetero and Fjord. I appreciate the connections made that facilitated the theoretical improvement in training.

**Weaknesses:**

I am struggling to understand in the paper what happens in the case of device heterogeneity - it is not explicitly described in the paper nor seems to be discussed sufficiently. I also lack visibility on what happens in case a node drops out from the computation or struggles to keep up with the load. Finally, it would be great to understand if the scheme introduces any bias towards the end result based on the proportional client datasets in the global scheme?

**Questions:**

I have a few questions regarding the manuscript, mainly:

- While the authors mention explicitly that both Hetero and Fjord are methods which are most similar to them they elected to only test against one of them. From my understanding Fjord came after Hetero with significant benefits. It would be great if this was compared against the proposed framework.
- What happens in the case a device drops from the computation?
- Most of the experiments are simulations of real-world performance, but only communication complexity is discussed as far as I can tell. It would be great to explicitly describe both the computation and memory complexity (benefits?) of the proposed method.